# Effects of Pyrroloquinoline Quinone on Lipid Metabolism and Anti-Oxidative Capacity in a High-Fat-Diet Metabolic Dysfunction-Associated Fatty Liver Disease Chick Model

**DOI:** 10.3390/ijms22031458

**Published:** 2021-02-01

**Authors:** Kai Qiu, Qin Zhao, Jing Wang, Guang-Hai Qi, Shu-Geng Wu, Hai-Jun Zhang

**Affiliations:** Risk Assessment Laboratory of Feed Derived Factors to Animal Product Quality Safety of Ministry of Agriculture & Rural Affairs & National Engineering Research Center of Biological Feed, Feed Research Institute, Chinese Academy of Agricultural Sciences, Beijing 100081, China; qiukai@caas.cn (K.Q.); zhaoqsmile@163.com (Q.Z.); wangjing@caas.cn (J.W.); qiguanghai@caas.cn (G.-H.Q.)

**Keywords:** pyrroloquinoline quinone, fatty liver syndrome, hepatocytes steatosis, oxidative stress, mitochondrial function, laying hens

## Abstract

Metabolic dysfunction-associated fatty liver disease (MAFLD) and its interaction with many metabolic pathways raises global public health concerns. This study aimed to determine the therapeutic effects of Pyrroloquinoline quinone (PQQ, provided by PQQ.Na_2_) on MAFLD in a chick model and primary chicken hepatocytes with a focus on lipid metabolism, anti-oxidative capacity, and mitochondrial biogenesis. The MAFLD chick model was established on laying hens by feeding them a high-energy low-protein (HELP) diet. Primary hepatocytes isolated from the liver of laying hens were induced for steatosis by free fatty acids (FFA) and for oxidative stress by hydrogen peroxide (H_2_O_2_). In the MAFLD chick model, the dietary supplementation of PQQ conspicuously ameliorated the negative effects of the HELP diet on liver biological functions, suppressed the progression of MAFLD mainly through enhanced lipid metabolism and protection of liver from oxidative injury. In the steatosis and oxidative stress cell models, PQQ functions in the improvement of the lipid metabolism and hepatocytes tolerance to fatty degradation and oxidative damage by enhancing mitochondrial biogenesis and then increasing the anti-oxidative activity and anti-apoptosis capacity. At both the cellular and individual levels, PQQ was demonstrated to exert protective effects of hepatocyte and liver from fat accumulation through the improvement of mitochondrial biogenesis and maintenance of redox homeostasis. The key findings of the present study provide an in-depth knowledge on the ameliorative effects of PQQ on the progression of fatty liver and its mechanism of action, thus providing a theoretical basis for the application of PQQ, as an effective nutrient, into the prevention of MAFLD.

## 1. Introduction

Metabolic dysfunction-associated fatty liver disease (MAFLD), previously known as non-alcoholic fatty liver disease (NAFLD), has been proposed as a better definition with “positive criteria” to identify the liver disease associated with known metabolic dysfunction, based on evidence of hepatic steatosis, in addition to one of the following three criteria, namely overweight/obesity, the presence of type 2 diabetes mellitus, or the evidence of metabolic dysregulation [1,2,3]. MAFLD has increasingly become a global public health issue, the prevalence of which continues to increase to affect up to one-fourth of the population with the incidences of obesity, dyslipidemia, diabetes, and insulin resistance being at their peak globally [4,5]. MAFLD is a complex disease with many interacting metabolic pathways, especially lipid metabolism [6]. Fatty liver develops when lipid acquisition exceeds lipid disposal, and pharmacotherapy of MAFLD is an unmet clinical need. Therefore, to enrich the knowledge of MAFLD etiology and its progresses and to explore nutrients that regulate lipid metabolism may be of great significance for the prevention and treatment of fatty liver.

The pathological progression of MAFLD is considered tentatively as a ‘three-hit’ process including steatosis, lipotoxicity, and inflammation [7]. Intracellular lipid accumulation results in a cascade of events in hepatocytes including lipotoxicity, oxidative stress, mitochondrial and endoplasmic reticulum dysfunction, and inflammation, and then induces hepatocyte death and fibrosis [8,9]. Mitochondrial beta-oxidation was increased as one of metabolic adaptations to restrain hepatic fat accumulation, but the progressive increased substrate delivery to mitochondrial respiratory chain activity during MAFLD impairs redox balance and decreases ATP synthesis [10,11]. All these factors are considered to be critical factors to initiate a vicious cycle of reducing the tolerance of hepatocytes towards damaging hits and aggravating MAFLD [12,13,14]. Redox imbalance including increased concentrations of reactive oxygen species (ROS) and impaired antioxidant defense system have been suggested to be highly associated to MAFLD pathogenesis [15]. Oxidative stress induced by ROS and inflammation are likely significant mechanisms that induce hepatic cell death and tissue injury [16]. Therefore, targeting to attenuate or halt mitochondrial dysfunction and redox imbalance which causes hepatocellular injury has been widely proposed as a novel and effective treatment strategy for MAFLD [17,18].

During MAFLD, steatosis is a reversible condition that can progress towards more severe liver damages such as steatohepatitis, fibrosis and cirrhosis if not halted or disrupted. Pyrroloquinoline quinone (PQQ), an aromatic tricyclic o-quinone, was first discovered as the third redox cofactor after nicotinamide and flavin in bacteria [19], and later found to be present in various plant and animal cells and many foods and biological fluids, such as milk, at pM to nM levels [20]. PQQ has a wide range of nutritional functions, such as promoting microbial and plant growth and being an essential nutrient for animals and effecting the growth, reproduction, and immune system. PQQ not only serves to mediate redox reactions in the mitochondrial respiratory chain, but also plays a potential role of scavenging ROS and attenuating oxidative stress in mitochondria and stimulating mitochondrial biogenesis through significant elevations in peroxisome proliferator-activated receptor-γ coactivator-1α (PGC-1α) protein expression [21,22,23]. Previous reports have demonstrated that PQQ deficiency influences normal mitochondrial, lipid, and energy metabolism in rats [24]. Early PQQ supplementation has persistent long-term protective effects on developmental programming of hepatic lipotoxicity and inflammation in obese mice [25]. Therefore, we hypothesized that PQQ would be a very promising therapeutic agent to reverse the progression of MAFLD as a new kind of antioxidant and mitochondrial nutrient but remains to be elucidated.

Besides humans, several species suffer adverse effects of the MAFLD, including chicken, cow, and cat. Laying hens appear to develop fatty livers under similar metabolic conditions of excess energy with humans and comparative physiology of numerous endotherms demonstrated that the regulation of de novo lipogenesis in the liver was very similar in humans and chicken, which makes the laying hen an appealing animal model widely used for investigating MAFLD and hepatic steatosis in humans [26,27,28,29]. In the current study, we sought to utilize laying hens as a more physiologically relevant preclinical model to determine the effectiveness of PQQ to attenuate the progression of MAFLD from the perspective of lipid metabolism, anti-oxidative capacity, and mitochondrial biogenesis.

## 2. Results

### 2.1. Effects of PQQ.Na_2_ on Performance and Egg Quality of Laying Hens

The performance and egg quality of laying hens were shown in Table 1. As for performance, the high-energy low-protein (HELP, metabolic energy (ME) = 12.75 MJ/Kg, crude protein (CP) = 13.0%) diet significantly (*p* < 0.05) increased average daily feed intake and the ratio of feed to egg, while no effect (*p* > 0.05) was observed on the laying rate and average egg weight in comparison to the control group (normal diet, ME = 11.03 MJ/Kg, CP = 16.2%). The supplementation of PQQ.Na_2_ on the basis of the HELP diet significantly reduced (*p* < 0.05) average daily feed intake of laying hens as compared with those fed the HELP diet. The negative effects of the HELP diet on the ratio of feed and egg were obviously ameliorated by the dietary supplementation of PQQ.Na_2_. With regard to egg quality, the HELP diet significantly decreased albumen height, yolk color, and Haugh unit, while showed no influences on the eggshell strength, eggshell thickness, and egg shape, as compared with the control group. Relative to the control group, the supplementation of PQQ.Na_2_ in the HELP diet could drastically (*p* < 0.05) recover its bad influences on albumen height and Haugh unit, and partly restore yolk color.

### 2.2. Effects of Dietary PQQ.Na_2_ on Liver Function of Laying Hens

In Figure 1A, livers of birds fed the HELP diet showed a significant increase in size and conspicuous color dodge toward to white from fuchsia relative to the control group, which could be seen by the naked eye directly. As compared with the control group, the crude fat content of liver tissue was significantly (*p* < 0.05) increased by the HELP diet (Figure 1B). The supplementation of PQQ.Na_2_ in the HELP diet effectively reduced (*p* < 0.05) the crude fat content of liver to the control level (Figure 1B) and reinstated the appearance of the liver to its normal size and color (Figure 1A).

In Figure 2A, pathological observations of liver tissues stained with hematoxylin–eosin (H&E) show that the number of fat vacuoles in livers of hens fed the HELP diet was significantly (*p* < 0.05) increased relative to the control group. The supplementation of PQQ.Na_2_ in the HELP diet effectively (*p* < 0.05) reduced the fat vacuoles number in livers. The collagen fiber of liver tissues was visualized by Masson staining (Figure 2B). The NAFLD activity score (NAS) of pathological sections of liver tissues in the HELP group was significantly (*p* < 0.05) greater than the control group, while was dose-dependently decreased (*p* < 0.05) by the supplementation of PQQ.Na_2_ (Figure 2C). The quantitative results of collagen fiber area of liver tissues showed that the supplementation of PQQ.Na_2_ could in a dose-dependent manner suppress liver fibrosis caused by the HELP diet (*p* < 0.05, Figure 2D). As shown in Figure 2E–H, the HELP diet significantly (*p* < 0.05) increased the content of triglyceride (TG), total cholesterol (TC), and malondialdehyde (MDA), and decreased the total superoxide dismutase (T-SOD) activity in livers of hens relative to the control diet. The addition of PQQ.Na_2_ obviously (*p* < 0.05) masked off the effects of the HELP diet on the content of TG, TC, and MDA, and the activity of T-SOD in livers of hens. The effect of supplementation of high concentration of PQQ.Na_2_ (0.16 mg/kg) in the HELP diet on decreasing the content of TC in liver is not as good as that of low concentration (0.08 mg/kg). As for mitochondrial function (Figure 2I,J), the citrate synthase (CS) and Cytochrome C oxidase (CCO) activity in liver of hens was significantly reduced (*p* < 0.05) by the HELP diet relative to the control diet. The supplementation of PQQ.Na_2_ in the HELP diet obviously (*p* < 0.05) enhanced the activity of CS and CCO in liver, while the effects of 0.16 mg/kg PQQ.Na_2_ in diet on the CS activity was smaller than those of 0.08 mg/kg.

The effects of dietary PQQ.Na_2_ on serum lipid metabolism and the anti-oxidative capacity of laying hens with fatty liver are shown in Table 2. Compared with the control group, hens fed the HELP diet had higher concentration of TG, TC, MDA, and low-density lipoprotein cholesterol (LDL-C), and activity of alanine transaminase (ALT), cholinesterase (ChE), and lactate dehydrogenase (LDH), and lower SOD activity in serum (*p* < 0.05). The supplementation of PQQ.Na_2_ in the HELP diet significantly decreased (*p* < 0.05) the content of TG, TC, and the activity of ALT, ChE and LDH, and increased (*p* < 0.05) SOD activity in serum, while did not reduce ChE activity. The addition of low dose of PQQ.Na_2_ not the high one in the HELP diet showed obvious effects on the reduction of MDA content (*p* < 0.05). In addition, the HELP diet with or without the supplementation of PQQ.Na_2_ had no influences on the aspartate aminotransferase (AST) activity and high-density lipoprotein cholesterol (HDL-C) content in serum relative to the control group.

### 2.3. Effects of PQQ.Na_2_ on the FFA-Induced Steatosis of Primary Hepatocytes

Compared with the control, the supplementation of PQQ.Na_2_ at the dose of 1, 2, 3, or 4 μmol/L in the culture medium had obvious cytotoxicity and significantly decreased (*p* < 0.05) the cell viability of primary hepatocytes after 24 h cultivation (Appendix A). The viability and morphology of primary hepatocytes cultured for 24 h in the medium with supplementation of 50, 100, 200, or 400 nmol/L PQQ.Na_2_ were not influenced relative to the control, and the 200 nmol/L PQQ.Na_2_ increased (*p* < 0.05) their viability (Appendix A). Therefore, doses of 0, 50, 100, and 200 nmol/L PQQ.Na_2_ without adverse effects on the growth of primary hepatocytes were chosen for subsequent tests. As shown in Appendix A, with the increase of free fatty acid (FFA) concentration in medium from 0, 0.5, 1, to 2 mmol/L, more and more lipid droplets stained by Oil Red O are formed in primary hepatocytes. The TG content was also quantified to be significantly increased (*p* < 0.05) in primary hepatocytes with the increasing concentration of FFA (Appendix A). Once the concentration of FFA was up to 2 mmol/L in medium, the viability of primary hepatocytes was significantly and sharply decreased (*p* < 0.01, data not shown). Therefore, 1 mmol/L FFA was used to induce hepatocyte steatosis in the following experiment.

Compared with the control, PQQ.Na_2_ supplementation partly and significantly masked (*p* < 0.05) the decreased (*p* < 0.05) cell viability caused by FFA induction (Figure 3A). As for lipid metabolism and anti-oxidative capacity, the increasing (*p* < 0.05) caused by FFA induction in the content of TG and MDA, and activity of ALT, AST, and gamma-glutamyl transpeptidase (GGT) relative to the control were dose-dependently reduced (*p* < 0.05) by PQQ.Na_2_ (Figure 3B–F). For mitochondrial function, FFA induction significantly lowered (*p* < 0.05) the content of mtDNA and the mRNA expression of genes regulated to mitochondrial metabolism and biogenesis including *TFAM*, *PGC-1α*, and *NRF-1*, which were totally or partly restored (*p* < 0.05) by the addition of PQQ.Na_2_ (Figure 3G–J). Regarding apoptosis signals, FFA induction significantly (*p* < 0.05) declined the mRNA expression of *Bax* and raised that of *Bcl-2* (Figure 3K–L). Relative to the control, PQQ.Na_2_ could not restore the decreased level of *Bax* expression caused by FFA induction, and the 100 nmol/L PQQ.Na_2_ even further decreased it (*p* < 0.05). The supplementation of 100 and 200 not 50 nmol/L PQQ.Na_2_ significant suppressed (*p* < 0.05) the FFA induced increase of the *Bcl-2* mRNA expression, even lower than the control (*p* < 0.05).

### 2.4. Effects of PQQ.Na_2_ on the Primary Hepatocytes upon Oxidative Stress

Primary hepatocytes pre-incubated with PQQ.Na_2_ showed obvious resistance (*p* < 0.05) to hydrogen peroxide (H_2_O_2_) induced decrease of cell viability, as compared with the control (Figure 4A). In the aspect of lipid metabolism and anti-oxidative capacity, relative to the control, the pre-treatment with PQQ.Na_2_ helped primary hepatocytes totally or partly (*p* < 0.05) avoid the H_2_O_2_ induced an increase (*p* < 0.05) of the content of ROS and MDA and a decrease (*p* < 0.05) of glutathione (GSH) content and the activity of T-SOD and catalase (CAT) (Figure 4B–F). For mitochondrial function, H_2_O_2_ induction significantly reduced (*p* < 0.05) mtDNA content and mRNA expression of *TFAM*, *PGC-1α*, *UCP-1*, and *NRF-1/2*. Upon H_2_O_2_ induction, pre-incubation with PQQ.Na_2_ basically dose-dependently increased (*p* < 0.05) mtDNA content, mRNA expression of *TFAM*, *PGC-1α*, *UCP-1*, and *NRF-1/2*, and ATPase activity in primary hepatocytes, even up to comparable levels to the control (Figure 4G–M). About apoptosis signals, PQQ.Na_2_ preconditioning significantly recovered (*p* < 0.05) the decreased (*p* < 0.05) mRNA expression of *Bax* and the increased (*p* < 0.05) mRNA expression of *Bcl-2* in primary hepatocytes upon H_2_O_2_ induction. Relative to the control, the mRNA expression of *Caspase-3* was not influenced by H_2_O_2_ stimulation, while pre-incubation with PQQ.Na_2_ dose-dependently decreased (*p* < 0.05) the mRNA expression of *Caspase-3* in primary hepatocytes (Figure 4N–P).

## 3. Discussion

MAFLD has become a global public health concern and there is no specific pharmacological treatment at present [4,5,30]. Lifestyle changes that promote weight loss, such as regular exercise, are considered as effective solutions to prevent and treat MAFLD; however, they are often difficult to maintain in the long term. PQQ was demonstrated to be closely related to mitochondrial, lipid, and energy metabolism [24] and has protective effects on hepatic lipotoxicity and inflammation in obese mice [25]. However, the actual application effects of PQQ on MAFLD and its precise mechanism remains unknown.

Recently, the MAFLD chick model constructed by the induction of a HELP diet was widely used for the studies of fatty liver diseases [31,32,33]. For laying hens, hepatic lipogenesis is enhanced due to the fatty acid requirement for egg production [34]. In the present study, we generated a model of MAFLD using laying hens fed a HELP diet. PQQ was proposed as a nutritionally important growth factor to improve reproduction performance in BALB/c mice and stimulated neonatal growth. The apparent requirement for PQQ for optimal growth of surviving neonates was estimated to be > or =0.3 mg PQQ/kg chemically defined diet [35]. Besides, in our previous study, the diet containing 0.08 or 0.12 mg/kg PQQ was found to reserve obvious protective effects against liver damage of laying hens induced by oxidized oil without effects on the performance [36]. Therefore, based on the previous data, the concentration of PQQ in diets in the current study was set as 0.08 and 0.16 mg/kg, which was hypothesized here as a nutritional strategy to prevent the HELP-induced MAFLD. TG accumulation in the liver is often used as an indicator of lipid metabolic disorders [37]. The liver tissues of hens fed the HELP diet showed significantly higher crude fat and TG content and more fat vacuoles relative to the control group, which indicated that the MAFLD model was successfully established. The liver is the central organ that controls lipid homeostasis, especially for laying hens with high demands of egg production [38]. The results agree with previous studies [39,40,41] that the MAFLD sharply decreased the egg production efficiency and egg quality. The disordered lipid metabolism in the MAFLD model was effectively reversed by the intake of dietary PQQ.Na_2,_ as evidenced by the decreased crude fat and TG content and fat vacuoles number in the liver tissue and decreased TG content in serum. PQQ was identified as a cofactor for mediating electron transfer in various enzymes including methanol dehydrogenase, glucose dehydrogenase, alcohol dehydrogenase, and aldehyde dehydrogenase [42,43]. Previous studies have evidenced that PQQ contributes to damage repair and delays cell senescence [44,45]. Therefore, we speculate that PQQ alleviated lipid metabolic dysfunction probably through regulating enzymes activity and increasing the tolerance of hepatocytes.

The nutrient PQQ was discovered as an antioxidant to be beneficial for health via helping the body resist harmful oxidative stress and abating inflammatory cells as well as inflammatory cytokine infiltration [46,47,48]. The activity of serum ALT and AST is an especially useful biomarker for detecting liver injury [49,50]. Moreover, the increase of serum LDH activity is another indicator of hepatocyte damage and loss of functional integrity. Mitochondrial function closely related to electron transport and ATP production can be partly assessed via the activity of oxidative enzymes such as CS and CCO. In the current study, in the MAFLD model, dietary PQQ significantly reduced the content of TG, TC, and LDL, and the activity of ALT and LDH in serum. Meanwhile, in both serum and liver, the activity of antioxidant enzyme, SOD, was increased and the content of MDA, an indicator of lipid peroxidation, was decreased. In addition, the activity of CS and CCO was also decreased in liver. Coincidentally, oxidative stress, hepatocyte inflammation, and apoptosis were demonstrated to occur simultaneously in the course of the MAFLD development, and drive the aggravation of MAFLD from early stage, steatosis, to advanced stage, cirrhosis [6,7,9]. Therefore, the above results indicated that PQQ protects liver from injury in MAFLD model probably through improving lipid metabolism, anti-oxidative activity, and mitochondrial function in the liver.

Steatosis is an essential step during pathological progression of MAFLD [7]. In the current study, the steatosis of liver tissues evaluated by the NAS score was significantly induced in the MAFLD model, dietary PQQ effectively relieved the steatosis of liver. In order to investigate the underlying mechanism of the therapeutic benefit of PQQ, primary hepatocytes were isolated from the liver of laying hens, and was used to induce steatosis by unsaturated FFA, the mixture of oleic acid and palmitic acid, which are the major mediators of hepatic steatosis in patients with MAFLD [51]. Oxidative stress results in the disorder of liver lipid metabolism, which is also an important mechanism inducing liver related diseases [52]. In human renal tubular epithelial cells, PQQ was demonstrated to have protective effects against oxidative stress-induced cellular senescence and inflammation [53]. PQQ could ameliorate autophagy-dependent apoptosis via lysosome-mitochondria axis in vascular endothelial cells [54]. In the present study, the treatment of PQQ significantly restored cell viability reduced by FFA induction and decreased the increased TG content. Upon FFA induction, the product of lipid peroxidation of primary hepatocytes, MDA, one of common indexes to evaluate the degree of oxidative stress, was significantly increased, while was meliorated by the supplementation of PQQ. The activity of liver enzymes denoting liver injury [49,50] including ALT, AST, and GGT were observably increased upon FFA induction, while was suppressed by the PQQ treatment. Mitochondrial malfunction is related to the onset of many diseases, and mtDNA encodes vital respiratory machinery of mitochondria [55]. mtDNA content could be regarded as an indicator of mitochondria biosynthesis. In addition, PGC-1α, NRFs, and TFAM are considered as the main factors involved in mitochondrial biogenesis [56,57]. Both Bax and Bcl-2 belonging to the Bcl-2 family are characterized apoptosis-mediating factors, and based on their biologic functions, can be classified into apoptosis-promoting and apoptosis-inhibiting factors, respectively [58]. In the current study, the treatment of PQQ significantly relieved the negative effects of FFA induction on hepatocytes including the decrease of mtDNA content, the mRNA expression of TFAM, PGC-1α, and NRF-1, and the ratio of Bax and Bcl-2 mRNA expression. These results demonstrate that PQQ suppresses the progression of steatosis in liver tissues and primary hepatocytes probably through increasing mitochondrial biogenesis and then decreasing oxidative stress and apoptosis.

Oxidative stress results in apoptosis, and ROS undoubtedly play an important role in the etiology and its progresses of numerous chronic liver diseases [59]. Antioxidants can effectively inhibit the oxidation, reduce lipid peroxides, and balance the oxidation and anti-oxidation defense systems [60]. The activity of two antioxidant enzymes, CAT and SOD and the content of GSH, can reflect the ability of cells to remove ROS and the resistance against oxidative damage [61]. Increase in UCP-1 expression level suppresses the accumulation of lipids in liver [62]. Caspase-3, one of the most important biochemical markers of apoptosis, plays a vital role in pathological cell death of numerous human diseases. PQQ-secreting probiotic *Escherichia coli* Nissle 1917 was found to have the capacity to ameliorate ethanol-induced oxidative damage and hyperlipidemia in rats [63]. In the treatment of knee osteoarthritis and osteoporosis, PQQ was used to inhibit oxidative stress and cell senescence [64,65]. In HK-2cells, PQQ was also confirmed to mask the high glucose-induced oxidative stress and apoptosis effect [66]. In the present study, H_2_O_2_ were used to induce oxidative stress to verify the protective effects of PQQ on primary hepatocytes. ROS and MDA levels, indicators of oxidative stress status, were significantly increased but cell viability was decreased in primary hepatocytes exposed to H_2_O_2_. Against H_2_O_2_ stimulation, PQQ pre-treatment could obviously promote cell viability and relieve the hepatocytes of oxidative stress effect. Moreover, in primary hepatocytes triggered by H_2_O_2_, pre-incubation with PQQ effectively enhanced the lipid decomposing ability reflected by UCP-1 mRNA expression and UCP-1 activity, antioxidant capacity reflected by CAT and SOD activities, mitochondrial biogenesis reflected by the mtDNA content and mRNA expression of TFAM, PGC-1α, and anti-apoptosis capacity reflected by mRNA expressions of Bax, Bcl-2, and Caspase-3. Therefore, we deduced that PQQ increases the lipid metabolism and antioxidant capacity probably through enhancing mitochondrial biogenesis and inhibiting apoptotic signaling pathway. The positive effects of PQQ on mitochondrial function in this study are in line with the previous studies which reported that PQQ deficiency in mice and rats exhibits reduced respiratory quotient [67,68,69], and PQQ has been demonstrated to be involved in the regulation of cellular energy metabolism and mitochondrial biogenesis in vivo and in vitro studies [24,68,69,70,71,72]. However, the final effects of PQQ on mitochondrial respiration in vivo and in vitro need to be directly evaluated and its underlying mechanism needs further research.

In summary, in the MAFLD chick model induced by a high-fat-diet, contributions of PQQ on suppressing the progression of MAFLD were mainly from the amelioration of lipid metabolism and protection of liver from oxidative damage. In the steatosis and oxidative stress cell models, PQQ functions in the improvement of lipid metabolism and hepatocytes tolerance to fatty degradation and oxidative damage through enhancing mitochondrial biogenesis and then increasing anti-oxidative activity and anti-apoptosis capacity. At both cellular and individual levels, PQQ was demonstrated to exert protective effects of hepatocyte and liver from fat accumulation through improvement of mitochondrial biogenesis and maintenance of redox homeostasis. The conclusions enrich the knowledge about effects of PQQ on the progression of fatty liver and its action mechanism and may provide a theoretical basis for the application of PQQ as an effective nutrient, into the prevention of MAFLD.

## 4. Materials and Methods

### 4.1. Ethics Statement

The animal experiment in the current study was conducted in accordance with the procedures approved (No: AEC-CAAS-20200107, Date: 7 January 2020) by the Animal Care and Use Committee of the Feed Research Institute of the Chinese Academy of Agricultural Sciences, Beijing.

### 4.2. Bird Management and Diet Allocation

A total of two hundred and eighty-eight 29-week-old Hy-line Brown laying hens with initial egg production rate of 93.3 ± 0.6% were randomly allotted into 1 of 4 dietary treatments. Each treatment had 6 replicates and each replicate contained 12 hens. The animal housing and handling procedures during the experiment were strictly in accordance with the requirements and recommendations of Hy-Line International Online Management Guide. Laying hens were fed a regular laying hen diet, which was adequate in all nutrients, prior to the feeding of the experimental diets. All birds had free access to water and diets during the test period. Feed intake, laying rate, egg weight of each replication was recorded weekly.

The composition and nutrition content of normal diet (the control, ME = 11.03 MJ/Kg, CP = 16.2%) and HELP diet (ME = 12.75 MJ/Kg, CP = 13.0%) were shown in Appendix A. In addition, on the basis of the HELP diet, PQQ.Na_2_ was supplemented at the concentration of 0.08 or 0.16 mg/kg diet. Therefore, four experimental diets in total were prepared and used in the animal experiment. All experimental diets provided vitamins and minerals for birds to meet or exceed the nutritional requirements.

### 4.3. Sample Collection of Animal Experiment

At the end of the 4-week trial, ten eggs were randomly selected from each replicate for egg quality determination. Besides, one bird that closed to the average BW of the replication was selected for sampling, and 6 birds were sampled per treatment. Firstly, the blood was collected from the wing vein and subsequently centrifuged at 3000× *g* for 15 min. The serum was harvested and stored at −20 °C for subsequent analysis. Secondly, liver organs were taken out and recorded by photographs. Two tissue samples from one Liver were collected, and one was fixed in 4% neutral buffered paraformaldehyde, the other one was homogenized at 4 °C for further component analysis.

### 4.4. Histological Analysis of the Liver

Fixed liver tissues were dehydrated and then embedded in paraffin. Carved wax blocks were cut into serial 4-mm-thick sections using a slicer (A550, MEDITE, Burgdorf, Germany). Then, the sections were dewaxed and stained with H&E. Five pathological sections per sample were observed using a light microscope (CK-40, Olympus, Tokyo, Japan) at 400× magnification. The representative sections with the average level in each treatment were selected for histological comparison. The NAS score, a semi-quantitative evaluation, was conducted by a pathologist based on the levels of hepatocyte steatosis, Inflammation within the lobules and ballooning degeneration of hepatocyte according to the guidelines for the diagnosis and treatment of NAFLD (Revised Edition, 2010) published by the Fatty Liver and Alcoholic Liver Disease Group of the Chinese Society of Liver Diseases. On additional unstained sections, Masson staining was used for the assessment of collagen deposition. Five sections per sample were scanned by use of a microscope (Eclipse Ci-L, Nikon, Japan) at 200 or 400× magnification. The collagen content on the digitalized images was quantified by using the built-in positive pixel counting algorithm in the Image-Pro Plus 6.0 (Media Cybemetics, Baltimore, MD, USA). All the histological measurements were performed by an observer unaware of the dietary treatments.

### 4.5. Egg Quality Determination

Three locations on the surface of eggs (i.e., the air cell, equator, and sharp end) were selected for the measurements of eggshell thickness using an Eggshell Thickness Gauge (ESTG1, Orka Technology Ltd., Ramat Hasharon, Israel). The mean value of 3 measurements per egg was calculated to represent its eggshell thickness. The eggshell-breaking strength was measured by an Egg Force Reader (Orka Technology Ltd., Ramat Hasharon, Israel). Albumen height, Haugh unit, and yolk color were determined by an Egg Analyzer (Orka Technology Ltd., Ramat Hasharon, Israel).

### 4.6. Primary Chicken Hepatocytes Isolation and Culture

Primary chicken hepatocytes were isolated from the Hy-line Brown laying hens (29-week old) using a two-step collagenase perfusion technique reported by Fraslin et al. [73] with some modifications. Briefly, after fasting for 12 h, hens were pre-anesthetized by the wing vein injection of heparin sodium 5 mL (7.5 mg/mL) and then anesthetized by an intraperitoneal injection of pentobarbital sodium (40 mg/kg). The abdominal cavity was opened, ribs were severed, and the breastbone was raised and maintained to permit easily access to the liver. A catheter (Venocath, Abbott, Sligo, Ireland) was inserted into the portal vein through the pancreaticoduodenal vein, which is much more accessible than the mesenteric one. The liver was first perfused for 15 min with 500 mL calcium-free HEPES (Sigma-Aldrich, Louis, MO, USA) buffer (pH 7.5), then for 30 min with the same buffer (500 mL) containing 100 mg collagenase (type A, Boehringer, Mannhelm, Germany) and 300 mg calcium chloride. Subsequently, the liver was immediately excised and transferred into a biological safety cabinet (HF-1100, Heal Force, Beijing, China). After the liver being washed three times by sterilized PBS, hepatocyte suspension in the liver was collected by disrupting the Glisson’s capsule with forceps and a blunt spatula. Then, the primary hepatocytes were cultured in Leibovitz medium (Gibco-BRL, Carlsbad, CA, USA) in a humidified CO_2_ incubator (HF90, Heal Force, Hongkong, China) maintaining 5% CO_2_ at 37 °C. Chemical reagents used for cell culture were purchased from Sigma-Aldrich. The fetal bovine serum (FBS), Williams’s Medium E, and Collagenase IV were obtained from Gibco-BRL. The cell culture materials were purchased from Corning (Corning, New York, NY, USA).

### 4.7. Cell Treatment and Sample Collection

Primary hepatocytes were plated in 6-well plates at 1 × 10^6^ cells per well. Free fatty acids (FFA, the mixture of oleic acid and palmitic acid at the ratio of 2:1, c/c) was added into the culture media as the final concentration of 0, 0.5, 1, or 2 mmol/L to induce fatty degeneration of hepatocytes for 24 h. Then, the cells were evaluated for viability or fixed for Oil Red O staining. After co-treated with FFA (0 or 1 mmol/L) and PQQ.Na_2_ (0, 50, 100, 200 or 400 nmol/L) for 24 h, hepatocytes were performed for the subsequent analysis. According to the oxidative stress model of primary hepatocytes of laying hens induced by H_2_O_2_ established by our lab, after treated with PQQ.Na_2_ (0, 50, 100, 200, or 400 nmol/L) for 24 h, hepatocytes was induced for oxidative stress with 4 mmol/L H_2_O_2_ for 4 h before subsequent analysis. PQQ.Na_2_ with the purity more than 99.9%, produced by microbial fermentation, was purchased from Shanghai Medical Life Science Research Center Co. (Shanghai, China).

### 4.8. Cell Viability Assay

Cell viability was determined by quantitative colorimetric assay with the MTT (3-(4,5)-dimethylthiahiazo(-z-y1)-3,5-di-phenytetrazoliumromide) method. Firstly, MTT was dissolved in PBS at 5 mg/mL, and 20 μL of MTT solution was added into each well for 3-h incubation at 37 °C. After incubation, the medium with MTT was discarded, and 160 μL of DMSO were added to dissolve the formed crystals with 10-min shaking. Absorbance of culture medium was then analyzed by a microplate reader (Synergy, BioTek, VT, USA) at 450 nm. The relative cell viability was directly calculated by absorbance value.

### 4.9. Quantitative Real-Time PCR Analysis

Total RNA was extracted from collected cells using TRIZOL reagent (Invitrogen Corporation, Shanghai, China). Aliquots of RNA samples were subjected to electrophoresis with 1.4% agarose-formaldehyde gels stained with ethidium bromide to verify the integrity. Then, RNA samples were reverse transcribed into cDNA using a PrimeScript^TM^ RT reagent Kit with gDNA Eraser (TaKaRa, Japan). The total DNA was extracted from liver tissues or cell samples using a QIAamp DNA Mini Kit (Qiagen, Dusseldorf, Germany) for the measurement of relative content of mitochondria (mitochondrial-DNA/nuclear-DNA). Real-time PCR was performed in Mx3500P (Agilent, Sacramento, CA, USA). The standard curves for each target gene, and melting curves were performed to insure a single specific PCR product for each gene. The information of primers was shown in Appendix A. Chicken β-actin mRNA was used as a reference gene for normalization purpose. PCR efficiency was confirmed based on the slope of cDNA relative standard curve that was produced using pooled samples. All samples were included in the same run of RT-PCR and repeated in triplicates. The 2^−ΔΔCt^ method was used to analyze the real-time RT-PCR data.

### 4.10. Chemical Analysis

Corresponding pretreatments were conducted for samples before chemical analysis. As for liver tissues, a total of 9 times the volume of 0.86% normal saline was added in the homogenization tube containing 0.1–0.2 g tissue to homogenize and centrifuge for 15 min (about 2500 rpm), and then the supernatant was collected ready for subsequent analysis. The serum samples were directly taken for index determination. As for cell samples, a total of 0.5 mL PBS (0.1 mol/L, pH 7–7.4) were added to the cell pellet and mixed. The cell suspension was sonicated 3~5 times under an ice-water bath with an interval of 30 s (3~5 s/time). Finally, the crushed homogenate was taken (no need to centrifuge) for subsequent analysis.

According to previous reports [74,75,76], the contents of TG and MDA in the liver tissue, serum and hepatocytes were determined using the corresponding kit with catalogue no. A110-1-1 and A003-1-2, respectively. The TC level in the liver tissue and serum was measured with the kit (no. A111-1-1). The activities of T-SOD, CS, and CCO in the liver tissue were analyzed with the corresponding kits (catalogue no. A001-1-2, A108-1-1, and A090-1-1, respectively). The activities of ALT, AST, and SOD in serum and hepatocytes were detected using the corresponding kits with catalogue no. C009-2-1, C010-2-1, and A001-3-2, respectively. The contents of HDL-C and LDL-C, and the activities of ChE and LDH were measured with the corresponding kits (A112-1-1, A113-1-1, A023-2-1, and A020-2-2, respectively). The ROS and GSH levels and the activities of GGT and CAT were determined using the corresponding kits with catalogue no. E004-1-1, A061-1-1, C017-1-1, and A007-1-1, respectively. The above commercial kits were purchased from Nanjing Jiancheng Bioengineering Institute (Nanjing, China). The protein concentrations of cell lysates were determined using a BCA Protein Assay Kit (Huaxingbio Science, Beijing, China). All the procedures were conducted strictly in accordance with the operation instructions of kits.

### 4.11. Statistical Analysis

The experimental data were subjected to one-way ANOVA procedure of SAS 9.1 (SAS Inst. Inc., Cary, NC, USA) and Tukey’s multiple rage test to determine the differences between treatments. Differences were considered significant at *p* ≤ 0.05. The data were presented as means ± SEM in Figures and as means together with a pooled SEM in Tables.

## Figures and Tables

**Figure 1 ijms-22-01458-f001:**
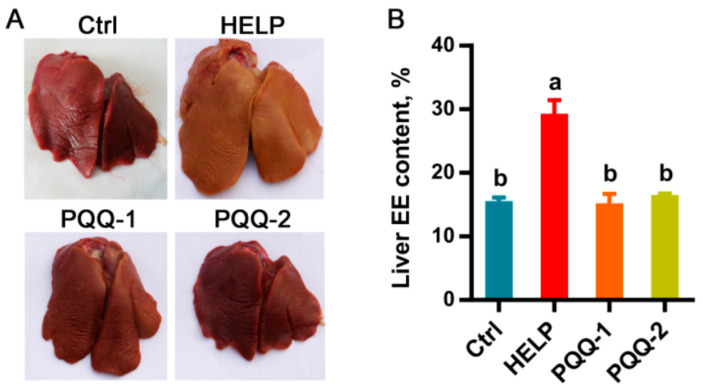
Effects of dietary PQQ.Na_2_ on fatty liver of laying hens. (**A**) Representative photographs of livers of experimental chicken. (**B**) The crude fat (ether extract, EE) content of liver tissues. PQQ.Na_2_, Pyrroloquinoline quinone disodium; Ctrl, control group, chickens fed the normal diet; HELP, chicken fed the high-energy low-protein diet; PQQ-1 and 2, chicken fed the high-energy low-protein diet supplemented with 0.08 and 0.16 mg/kg PQQ.Na_2_, respectively. Bars without the same small letters mean significant difference (*p* < 0.05).

**Figure 2 ijms-22-01458-f002:**
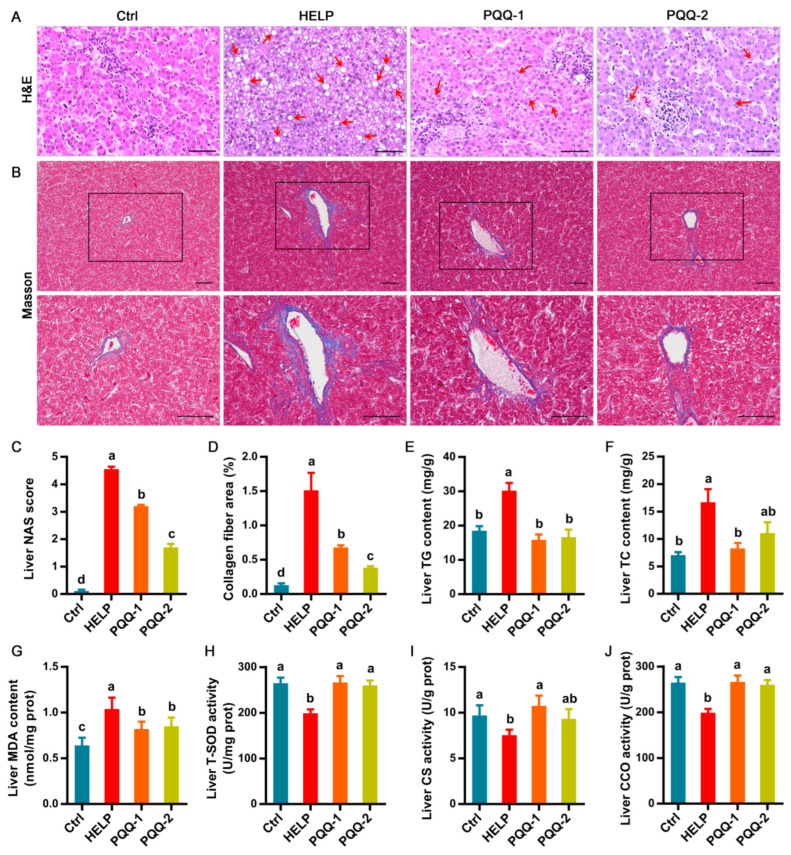
Effects of dietary PQQ.Na_2_ on liver functions of laying hens with fatty liver. (**A**) Histopathological observations (H&E staining) of liver tissues (bar, 100 μm). (**B**) Collagen fiber imaging (Masson staining) of liver tissues (bar, 250 μm). (**C**) Assessment of non-alcoholic fatty liver disease (NAFLD) activity score (NAS) of liver tissues based on histological sections. (**D**) The collagen fiber area of liver tissues. (**E**) The triglyceride (TG) content of liver tissues. (**F**) The total cholesterol (TC) content of liver tissues. (**G**) The malondialdehyde (MDA) content of liver tissues. (**H**) The total superoxide dismutase (T-SOD) activity in liver tissues. (**I**) The citrate synthase (CS) activity in liver. (**J**) The Cytochrome C oxidase (CCO) activity in liver tissues. PQQ.Na_2_, Pyrroloquinoline quinone disodium; Ctrl, control group, chickens fed the normal diet; HELP, chicken fed the high-energy low-protein diet; PQQ-1 and 2, chicken fed the high-energy low-protein diet supplemented with 0.08 and 0.16 mg/kg PQQ.Na_2_, respectively. Bars without the same small letters mean significant difference (*p* < 0.05).

**Figure 3 ijms-22-01458-f003:**
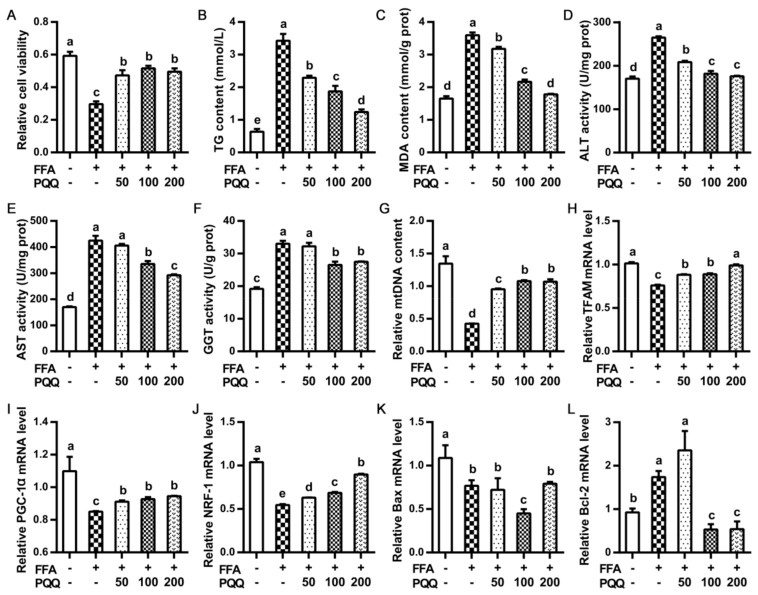
Effects of PQQ.Na_2_ on lipid metabolism, anti-oxidative capacity, hepatic mitochondrial function, and apoptosis signals of primary hepatocytes treated with free fatty acid. (**A**) The relative cell viability. (**B**) The triglyceride (TG) content of hepatocytes. (**C**) The malondialdehyde (MDA) content of hepatocytes. (**D**) The alanine transaminase (ALT) activity in hepatocytes. (**E**) The Aspartate aminotransferase (AST) activity in hepatocytes. (**F**) The gamma-glutamyl transpeptidase (GGT) activity in hepatocytes. (**G**) The relative mtDNA content of hepatocytes. (**H**–**J**) The relative mRNA expressions of mitochondrial genes in hepatocytes, including *TFAM*, *PGC-1α*, and *NRF-1*. (**K**,**L**) The relative mRNA expressions of genes related to apoptosis in hepatocytes, including *Bax* and *Bcl-2*. Free fatty acid (FFA) was added at a concentration of 1 mmol/L. PQQ.Na_2_ (Pyrroloquinoline quinone disodium, PQQ) was added at a concentration of 50, 100, or 200 nmol/L. Bars without the same small letters mean significant difference (*p* < 0.05).

**Figure 4 ijms-22-01458-f004:**
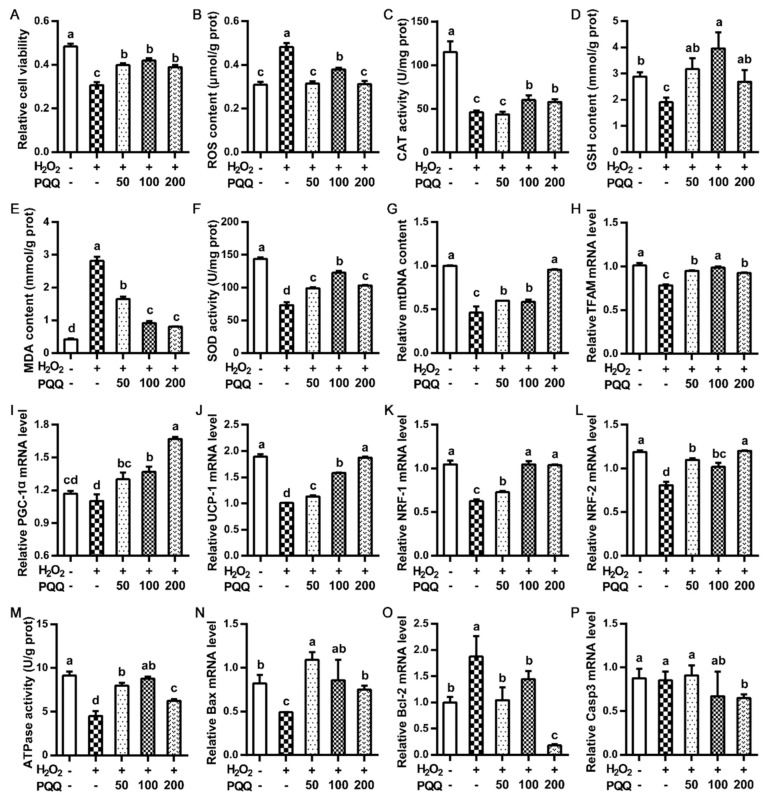
Effects of PQQ.Na_2_ on lipid metabolism, anti-oxidative capacity and hepatic mitochondrial function of primary hepatocytes treated with hydrogen peroxide. (**A**) The relative cell viability. (**B**) The Reactive oxygen species (ROS) content of hepatocytes. (**C**) The catalase (CAT) activity in hepatocytes. (**D**) The glutathione (GSH) content of hepatocytes. (**E**) The malondialdehyde (MDA) content of hepatocytes. (**F**) The superoxide dismutase (SOD) activity in hepatocytes. (**G**) The relative mtDNA content of hepatocytes. (**H**–**L**) The relative mRNA expressions of mitochondrial genes in hepatocytes, including *TFAM*, *PGC-1α*, *UCP-1*, *NRF-1*, and *NRF-2*. (**M**) The ATPase activity in hepatocytes. (**N**–**P**) the mRNA expression of genes related to apoptosis in hepatocytes, including *Bax*, *Bcl-2*, and *Caspase-3*. Hydrogen peroxide (H_2_O_2_) was added at a concentration of 4 mmol/L. PQQ.Na_2_ (Pyrroloquinoline quinone disodium, PQQ) was added at a concentration of 50, 100, or 200 nmol/L. Bars without the same small letters mean significant difference (*p* < 0.05).

**Table 1 ijms-22-01458-t001:** Effects of PQQ.Na_2_ on performance and egg quality of laying hens.

Items	Ctrl	HELP	PQQ-1	PQQ-2	SEM	*p*-Value
Performance						
Laying rate, %	91.02	91.04	91.36	92.71	0.58	0.71
Average egg weight, g	58.87	59.33	58.41	59.22	0.25	0.59
Average daily feed intake, g	117.72 ^b^	132.67 ^a^	118.20 ^b^	122.38 ^b^	1.99	0.03
Feed/egg	2.20 ^b^	2.43 ^a^	2.35 ^ab^	2.37 ^ab^	0.04	0.17
Egg quality						
Eggshell strength, N/m^2^	43.76	43.65	44.97	40.42	0.77	0.19
Eggshell thickness, mm	0.43	0.43	0.44	0.44	0.002	0.77
Egg shape index	1.30	1.30	1.31	1.30	0.005	0.86
Albumen height, mm	6.65 ^a^	5.64 ^b^	6.57 ^a^	6.51 ^a^	0.11	<0.01
Yolk color	6.36 ^a^	5.57 ^b^	5.78 ^ab^	5.89 ^ab^	0.12	0.02
Haugh unit	78.96 ^a^	72.44 ^b^	79.38 ^a^	78.26 ^a^	0.74	<0.01

PQQ.Na_2_, Pyrroloquinoline quinone disodium; Ctrl, control group, chickens fed the normal diet; HELP, chicken fed the high-energy low-protein diet; PQQ-1 and 2, chicken fed the high-energy low-protein diet supplemented with 0.08 and 0.16 mg/kg PQQ.Na_2_, respectively; SEM, standard error of mean. In the same row, values without the same small letter superscripts mean significant difference (*p* < 0.05).

**Table 2 ijms-22-01458-t002:** Effects of dietary PQQ.Na_2_ on serum lipid metabolism and anti-oxidative capacity of laying hens with fatty liver.

Items	Ctrl	HELP	PQQ-1	PQQ-2	SEM	*p*-Value
Triglyceride (TG), mmol/L	11.22 ^b^	15.38 ^a^	8.25 ^b^	10.63 ^b^	0.78	<0.01
Total cholesterol (TC), mmol/L	3.03 ^b^	3.94 ^a^	2.77 ^b^	3.19 ^b^	0.23	<0.01
Alanine transaminase (ALT), U/L	72.47 ^a^	84.66 ^a^	50.33 ^b^	54.26 ^b^	3.93	<0.01
Aspartate aminotransferase (AST), U/L	7.47	8.66	8.61	7.65	0.58	0.86
Cholinesterase (ChE), U/L	62.81 ^b^	85.69 ^a^	87.84 ^a^	78.75 ^a^	3.53	<0.01
Lactate dehydrogenase (LDH), U/μL	4.06 ^b^	5.13 ^a^	3.64 ^b^	3.46 ^b^	0.17	<0.01
Malondialdehyde (MDA), nmol/mL	6.40 ^b^	9.09 ^a^	6.74 ^b^	8.76 ^a^	0.37	<0.01
Superoxide dismutase (SOD), U/mL	274.49 ^a^	234.35 ^b^	274.33 ^a^	269.56 ^a^	6.21	0.04
High-density lipoprotein cholesterol (HDL-C), mmol/L	1.57	1.54	1.55	1.58	0.03	0.97
Low-density lipoprotein cholesterol (LDL-C), mmol/L	1.72 ^b^	2.68 ^a^	1.40 ^b^	1.61^b^	0.13	<0.01

PQQ.Na_2_, Pyrroloquinoline quinone disodium; Ctrl, control group, chickens fed the normal diet; HELP, chicken fed the high-energy low-protein diet; PQQ-1 and 2, chicken fed the high-energy low-protein diet supplemented with 0.08 and 0.16 mg/kg PQQ.Na_2_, respectively; SEM, standard error of mean. In the same row, values without the same small letter superscripts mean significant difference (*p* < 0.05).

## Data Availability

All data were shown in Tables and Figures in the main text or supplemental files. The original data are also available from corresponding authors by E-mail.

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
