# Peer review of "Effects of Pyrroloquinoline Quinone on Lipid Metabolism and Anti-Oxidative Capacity in a High-Fat-Diet Metabolic Dysfunction-Associated Fatty Liver Disease Chick Model"

_ijms, 2021, doi:10.3390/ijms22031458_

Round 1

Reviewer 1 Report

The authors have satisfactorily answered to the raised questions

Author Response

Thank you for your recognition.

Reviewer 2 Report

The paper, Effects of Pyrroloquinoline Quinine on Lipid Metabolism and Anti-Oxidative Capacity in a High-Fat-Diet Nonalcoholic Fatty Liver Disease Chick Model, has potential merit and adds to observations regarding possible attributes of PQQ exposure. In its current form, however, the paper needs extensive editing related to English usage.

An example of poor editing is found in the title. PQQ is a pyrroloquinoline quinone, not quinine, i.e., the natural cinchona alkaloid. The error, perhaps a misspelling, is noted throughout the paper. Further, the first sentence of the introduction contains over 50 words and conveys too much information. Throughout the manuscript, other long sentences are difficult to decipher. There are also numerous examples related to preposition and direct/indirect article usage that need correction. Please have someone with good English skills go through the manuscript.

On the positive side, the paper's organization is adequate, and the data provided support the article's assertions. However, the authors should provide their rationale for the dietary levels of PQQ used. Table 1 notes that 80 or 160 micrograms of sodium-PQQ were added per Kg of diet, which supported and improved growth. For comparison, neonatal rats' requirement is reported as 200-400 micrograms of sodium-PQQ per Kg of highly purified diets. These concentrations are in the lowest range of known vitamin requirements. The authors should comment on that point.

Further - According to a recently published USA GRAS notice - (http://www.fda.gov/Food/IngredientsPackagingLabeling/GRAS/NoticeInventory/default.htm), 

your diets probably contained at least 6 micrograms of PQQ from the corn added and 6 to 9 from the soybean added (expressed per Kg). PQQ imidazole derivatives are also a source of PQQ. Their levels are suggested to be ~10 times the reported values for PQQ when present in foods. With that said, you were perhaps only adding 2-3 times what was in the basal diet. Should the levels of PQQ that you used in diets be viewed as pharmacologic?  The question is important in that it speaks to the issue of nutritional versus pharmacologic importance.

Author Response

Question 1: The paper, Effects of Pyrroloquinoline Quinine on Lipid Metabolism and Anti-Oxidative Capacity in a High-Fat-Diet Nonalcoholic Fatty Liver Disease Chick Model, has potential merit and adds to observations regarding possible attributes of PQQ exposure. In its current form, however, the paper needs extensive editing related to English usage.

An example of poor editing is found in the title. PQQ is a pyrroloquinoline quinone, not quinine, i.e., the natural cinchona alkaloid. The error, perhaps a misspelling, is noted throughout the paper. Further, the first sentence of the introduction contains over 50 words and conveys too much information. Throughout the manuscript, other long sentences are difficult to decipher. There are also numerous examples related to preposition and direct/indirect article usage that need correction. Please have someone with good English skills go through the manuscript.

Answer: We feel so sorry for the mistakes in the manuscript. According to your suggestion, we have found an English native speaker, one international PhD student, to help us revise the manuscript.

Question 2: On the positive side, the paper's organization is adequate, and the data provided support the article's assertions. However, the authors should provide their rationale for the dietary levels of PQQ used. Table 1 notes that 80 or 160 micrograms of sodium-PQQ were added per Kg of diet, which supported and improved growth. For comparison, neonatal rats' requirement is reported as 200-400 micrograms of sodium-PQQ per Kg of highly purified diets. These concentrations are in the lowest range of known vitamin requirements. The authors should comment on that point.

Further - According to a recently published USA GRAS notice - (http://www.fda.gov/Food/IngredientsPackagingLabeling/GRAS/NoticeInventory/default.htm),

your diets probably contained at least 6 micrograms of PQQ from the corn added and 6 to 9 from the soybean added (expressed per Kg). PQQ imidazole derivatives are also a source of PQQ. Their levels are suggested to be ~10 times the reported values for PQQ when present in foods. With that said, you were perhaps only adding 2-3 times what was in the basal diet. Should the levels of PQQ that you used in diets be viewed as pharmacologic?  The question is important in that it speaks to the issue of nutritional versus pharmacologic importance.

Answer: Thanks for your valuable suggestion. We have improved our discussion in Line 268-276 as follows according to your suggestion. Indeed, 0.08 and 0.16 mg/kg PQQ per Kg of diet was set in the current study according to the results of our previous study about the effects of dietary PQQ on liver damage of laying hens induced by oxidized oil. In the discussion section, together with the previous study about the effects of PQQ on BALB/c mice, we supplemented to explain our rationale for the dietary levels of PQQ used. In the present study, dietary PQQ showed positive effects to prevent the occurrence or aggravation of MAFLD. It needs further research about whether PQQ could be used for pharmacologic treatment of MAFLD. We have corrected “the treatment of MAFLD” into “the prevention of MAFLD” in the Abstract and Discussion sections for better description of our results. We do hope we understand your questions correctly. Thanks for the kindly help for our improvement.

PQQ was proposed as a nutritionally important growth factor to improve reproduction performance in BALB/c mice and stimulated neonatal growth. The apparent requirement for PQQ for optimal growth of surviving neonates was estimated to be > or = 0.3 mg PQQ/kg chemically-defined diet [35]. Besides, in our previous study, the diet containing 0.08 or 0.12 mg/kg PQQ was found to reserve obvious protective effects against liver damage of laying hens induced by oxidized oil without effects on the performance [36]. Therefore, based on the previous data, the concentration of PQQ in diets in the current study was set as 0.08 and 0.16 mg/kg, which was hypothesized here as a nutritional strategy to prevent the HELP-induced MAFLD.

  1. Steinberg, F.M.; Gershwin, M.E.; Rucker, R.B. Dietary pyrroloquinoline quinone: growth and immune response in BALB/c mice. J. Nutr., 1994, 124, 744-753.
  2. Wang, J.; Zhang, H.J.; Xu, L.; Long, C.; Samuel, K.G.; Yue, H.Y.; Sun, L.L.; Wu, S.G.; Qi, G.H. Dietary supplementation of pyrroloquinoline quinone disodium protects against oxidative stress and liver damage in laying hens fed an oxidized sunflower oil-added diet. Animal, 2016, 10, 1129-1136.

Reviewer 3 Report

To the authors of the manuscript,

The work highlights the relevance of the compound PQQ in an animal model of laying hens and it demonstrates its effectivity in reducing steatosis and inflammation. However, I think there is still work to be realized before its publication so that I recommend to perform MAJOR REVISIONS priorly.

Please find attached a list of some concerns:

MAJOR

  • In the introduction the terms MAFLD/NAFLD must be completely clear. Indeed, in the paper PQQ is reversing NASH and you do not mention it. Please revise it properly.
  • The figures are really hard to follow as they are labelled with Trt1-3 and significances are hard to distinguish. If you indicated the tittle of each treatment and significances with * it would be much easier.
  • I do not fully understand the use of laying hens instead of mice for a pre-clinical study. I would appreciate, at least, the confirmation of the results with a mammal model more similar to humans. 
  • In Fig. 2C you measure the NAS score but you should indicate really well what you have based on for determining it. 
  • In the work, and particularly in the Discussion, you focus on other non-liver related experiments to highlight the role of inflammation and apoptosis. Furthermore, you already have results in your manuscript that support the role of such processes in the aggravation of NAFLD from benign stages (steatosis) to more severe ones (cirrhosis). This concern is also linked to the previous one, so that if you better clarify these concepts the text can be improved.

MINOR

  • Results in Fig. 2 should be regrouped in order to make their reading much cleared. The same happens with Fig. 4
  • I consider that Fig. 3 should be part of Supplementary material.

Author Response

MAJOR

Question 1: In the introduction the terms MAFLD/NAFLD must be completely clear. Indeed, in the paper PQQ is reversing NASH and you do not mention it. Please revise it properly.

Answer: As you suggested, we have detailed the terms MAFLD/NAFLD in the introduction in Line 39-44. In addition, we also supplemented related discussion about NASH in Line 306-312 and 334-336. The modifications made in main text are shown as follows.

Line 39-44: Metabolic dysfunction-associated fatty liver disease (MAFLD), previously known as non-alcoholic fatty liver disease (NAFLD), has been proposed as a better definition with “positive criteria” to identify the liver disease associated with known metabolic dysfunction, based on evidence of hepatic steatosis, in addition to one of the following three criteria, namely overweight/obesity, presence of type 2 diabetes mellitus, or evidence of metabolic dysregulation [1-3]. MAFLD has increasingly become a global public health issue, the prevalence of which continues to increase to affect up to one-fourth of the population with the worldwide surge in the incidence of obesity, dyslipidemia, diabetes, and insulin resistance [4-5].

Line 306-312: Steatosis is an essential step during pathological progression of MAFLD [7]. In the current study, the steatosis of liver tissues evaluated by the NAS score was significantly induced in the MAFLD model, dietary PQQ effectively relieved the steatosis of liver. In order to investigate the underlying mechanism of the therapeutic benefit of PQQ, primary hepatocytes were isolated from the liver of laying hens, and was used to induce steatosis by unsaturated FFA, the mixture of oleic acid and palmitic acid, which are the major mediators of hepatic steatosis in patients with MAFLD [49].
Line 334-336: These results demonstrate that PQQ suppresses the progression of steatosis in liver tissues and primary hepatocytes probably through increasing mitochondrial biogenesis and then decreasing oxidative stress and apoptosis.

Question 2: The figures are really hard to follow as they are labeled with Trt1-3 and significances are hard to distinguish. If you indicated the tittle of each treatment and significances with * it would be much easier.

Answer: We have re-labeled Trt1-3 as HELP, PQQ-1, and PQQ-2, respectively in all related tables and figures for easy reading. Owing to the up to 4 or 5 treatments in this study, using * to mark the significations between every two groups will looks cluttered. Thank you for your understanding and tolerance.

Question 3: I do not fully understand the use of laying hens instead of mice for a pre-clinical study. I would appreciate, at least, the confirmation of the results with a mammal model more similar to humans.

Answer: As you said, rodents are often used as animal models in the pre-clinical studies. However, MAFLD-induced by HELP diets in rodents does not completely relate to human MAFLD. Comparative physiology of numerous endotherms indicates that humans and chickens use the liver for ≥ 90% of de novo lipogenesis [1-2]. In rodents and rabbits, adipose tissues and livers contribute about equally to de novo lipogenesis [3]. In addition to in humans, excess hepatic lipid deposition is also a common pathology in poultry, contributed to by many factors, such as aging, overfeeding, diseases, genetic background, or toxins [4]. Therefore, some studies have suggested that chickens would be an appealing animal model for human MAFLD and hepatic steatosis [5-6]. We supplemented related explain in the Introduction section in Line 87-91 as follows.

Laying hens appear to develop fatty livers under similar metabolic conditions of excess energy with humans and comparative physiology of numerous endotherms demonstrated that regulation of de novo lipogenesis in the liver was very similar in humans and chicken, which makes the laying hen an appealing animal model widely used for investigating MAFLD and hepatic steatosis in humans [26-29].

[1] Galton DJ. Lipogenesis in human adipose tissue. J Lipid Res. 1968 Jan;9(1):19-26.

[2] Leveille GA, O'Hea EK, Chakbabarty K. In vivo lipogenesis in the domestic chicken. Proc Soc Exp Biol Med. 1968 Jun;128(2):398-401

[3] Laliotis GP, Bizelis I, Rogdakis E. Comparative Approach of the de novo Fatty Acid Synthesis (Lipogenesis) between Ruminant and Non Ruminant Mammalian Species: From Biochemical Level to the Main Regulatory Lipogenic Genes. Curr Genomics. 2010 May;11(3):168-183

[4] Fournier E, Peresson R, Guy G, Hermier D. Relationships between storage and secretion of hepatic lipids in two breeds of geese with different susceptibility to liver steatosis. Poult Sci. 1997 Apr;76(4):599-607.

[5] Ayala I, Castillo AM, Adánez G, Fernández-Rufete A, Pérez BG, Castells MT. Hyperlipidemic chicken as a model of non-alcoholic steatohepatitis. Exp Biol Med (Maywood). 2009 Jan;234(1):10-16.

[6] Makovicky P, Dudova M, Tumova E, Rajmon R, Vodkova Z. Experimental study of non-alcoholic fatty liver disease (NAFLD) on a model of starving chickens: is generalization of steatosis accompanied by fibrosis of the liver tissue? Pathol Res Pract. 2011 Mar 15;207(3):151-155.

Question 4: In Fig. 2C you measure the NAS score but you should indicate really well what you have based on for determining it.

Answer: According to your suggestion, we detailed the method used for NASH valuation in the Materials and Methods section in Line 413-414 as follows.

The NAS score, a semi-quantitative evaluation, was conducted by a pathologist based on the levels of hepatocyte steatosis, Inflammation within the lobules and ballooning degeneration of hepatocyte according to the guidelines for the diagnosis and treatment of NAFLD (Revised Edition, 2010) published by the Fatty Liver and Alcoholic Liver Disease Group of the Chinese Society of Liver Diseases.

Question 5: In the work, and particularly in the Discussion, you focus on other non-liver related experiments to highlight the role of inflammation and apoptosis. Furthermore, you already have results in your manuscript that support the role of such processes in the aggravation of NAFLD from benign stages (steatosis) to more severe ones (cirrhosis). This concern is also linked to the previous one, so that if you better clarify these concepts the text can be improved.

Answer: We have improved our discussion according to your suggestion in Line 300-305 as follows. As you said, oxidative stress, hepatocyte inflammation and apoptosis occur simultaneously in the course of MAFLD development and drive the aggravation of MAFLD from early stage, steatosis, to advanced stage, cirrhosis. In other non-liver related experiments, PQQ was found to play a role to against inflammation and apoptosis, which was verified again in our study. Therefore it is reasonable to deduce that PQQ protects liver from injury in MAFLD model probably through improving lipid metabolism, anti-oxidative activity, and mitochondrial function in the liver. Thanks for your constructive suggestion.

Coincidentally, oxidative stress, hepatocyte inflammation and apoptosis were demonstrated to occur simultaneously in the course of MAFLD development, and drive the aggravation of MAFLD from early stage, steatosis, to advanced stage, cirrhosis [6, 7, 9]. Therefore, the above results indicated that PQQ protects liver from injury in MAFLD model probably through improving lipid metabolism, anti-oxidative activity, and mitochondrial function in the liver.

MINOR

Question 6: Results in Fig. 2 should be regrouped in order to make their reading much cleared. The same happens with Fig. 4

Answer: The panels of Figure 2 and 4 were arranged in order as they referred in the Results section according to the requirements of Journal. I will modify it, if the editor considers it necessary. Thank you for your understanding and tolerance.

Question 7: I consider that Fig. 3 should be part of Supplementary material.

Answer: According to your suggestion, we have move Fig. 3 into the supplementary material and re-labeled as Fig. S1. The other Figures in the main text were re-labeled accordingly.

Round 2

Reviewer 3 Report

To the authors,

I appreciate the effort made as I have found answered all my concerns. Thus, I have recommended to ACCEPT it. 

Best regards.

This manuscript is a resubmission of an earlier submission. The following is a list of the peer review reports and author responses from that submission.

Round 1

Reviewer 1 Report

The authors aimed to evaluate the effect of PQQ supplementatin on laying hens fed a NAFLD inducing diet. The results are novel and in line with previously published reports on the potentially protective effect of PQQ on mitochondrial function. To sustain their conclusions they use an animal model and a cell culture model, both suggest that PQQ suplementation improves mitochondrial activity in animals or cells exposed to lipotoxicity. However, to substantiate the proposal the authors should directly evaluate mitochondrial respiration in the in vitro model and oxygen consumption rates in the hens. It would also be relevant to test it these effects are mediated by the activation of AMPK both in vitro and in vivo as has been previously proposed for PQQ activity as a positive regulator of PGC-1alpha levels. A minor issue, acronims should be defined first in the text and not in figures or figure legends, and the HELP diet should be described not only in methods but also in the results section for more clarity.

Author Response

The authors aimed to evaluate the effect of PQQ supplementatin on laying hens fed a NAFLD inducing diet. The results are novel and in line with previously published reports on the potentially protective effect of PQQ on mitochondrial function. To sustain their conclusions they use an animal model and a cell culture model, both suggest that PQQ suplementation improves mitochondrial activity in animals or cells exposed to lipotoxicity. However, to substantiate the proposal the authors should directly evaluate mitochondrial respiration in the in vitro model and oxygen consumption rates in the hens. It would also be relevant to test it these effects are mediated by the activation of AMPK both in vitro and in vivo as has been previously proposed for PQQ activity as a positive regulator of PGC-1alpha levels. A minor issue, acronyms should be defined first in the text and not in figures or figure legends, and the HELP diet should be described not only in methods but also in the results section for more clarity.

Response: Thanks for your recognition of our work and constructive suggestions. The effects on mitochondrial respiration of hepatocytes and oxygen consumption of individuals are belonging to the nutritional functions of PQQ, and have been studied. Nutritional studies have revealed that PQQ deficiency in mice and rats exhibits various systemic responses, including reduced respiratory quotient [1–3]. In animal models and cultured cells, PQQ has been demonstrated to involve the activation or expression of factors which play a central role in the regulation of cellular energy metabolism (e.g., promotion of β-oxidation and mitochondrial respiration) and mitochondrial biogenesis [4-10]. In this study, we mainly focus on the antioxidant role of PQQ and its prevention and treatment action on fatty liver. In fact, PQQ also has a wide range of nutritional functions, such as promoting microbial and plant growth, and being an essential nutrient for animals and effecting the growth, reproduction, and immune. We have supplemented related information in the introduction part for easy understanding. In addition, according the reviewer’s suggestion, we defined all acronyms with their full names when they first appear in the text.

[1] Killgore J, Smidt C, Duich L, et al. Nutritional importance of pyrroloquinoline quinone. Science, 1989; 245: 850-852.

[2] Steinberg FM, Gershwin ME, Rucker RB. Dietary pyrroloquinoline quinone: growth and immune response in BALB/c mice. J. Nutr., 1994; 124: 744-753.

[3] Steinberg F, Stites T, Anderson P, et al. Pyrroloquinoline quinone improves growth and reproductive performance in mice fed chemically defined diets. Exp. Biol. Med. 2003; 228: 160-166.

[4] W. Chowanadisai, K.A. Bauerly, E. Tchaparian, A. Wong, G.A. Cortopassi, R.B. Rucker. Pyrroloquinoline quinone stimulates mitochondrial biogenesis through cAMP response element-binding protein phosphorylation and increased PGC-1alpha expression. J. Biol. Chem., 2010; 285: 142-152

[5] E. Tchaparian, L. Marshal, G. Cutler, K. Bauerly, W. Chowanadisai, M. Satre, et al. Identification of transcriptional networks responding to pyrroloquinoline quinone dietary supplementation and their influence on thioredoxin expression, and the JAK/STAT and MAPK pathways. Biochem. J., 2010; 429: 515-526

[6] T. Stites, D. Storms, K. Bauerly, J. Mah, C. Harris, A. Fascetti, et al. Pyrroloquinoline quinone modulates mitochondrial quantity and function in mice. J. Nutr., 2006; 136: 390-396

[7] K. Bauerly, C. Harris, W. Chowanadisai, J. Graham, P.J. Havel, E. Tchaparian, et al. Altering pyrroloquinoline quinone nutritional status modulates mitochondrial, lipid, and energy metabolism in rats. PLoS One, 2011; 6: e21779

[8] F. Steinberg, T.E. Stites, P. Anderson, D. Storms, I. Chan, S. Eghbali, et al. Pyrroloquinoline quinone improves growth and reproductive performance in mice fed chemically defined diets. Exp. Biol. Med., 2003; 228: 160-166.

[9] F.M. Steinberg, M.E. Gershwin, R.B. Rucker.Dietary pyrroloquinoline quinone: growth and immune response in BALB/c mice. J. Nutr., 1994; 124: 744-753

[10] K.A. Bauerly, D.H. Storms, C.B. Harris, S. Hajizadeh, M.Y. Sun, C.P. Cheung, et al. Pyrroloquinoline quinone nutritional status alters lysine metabolism and modulates mitochondrial DNA content in the mouse and rat. Biochim. Biophys. Acta, 2006; 1760: 1741-1748.

Reviewer 2 Report

NAFLD prognosis is depent on NASH developement and liver fibrosis progression. I miss hystological analysis of the liver to determine inflammatory acitvity -such as NASH SAF score in humans.

The diet  period was too short. Only one months. Control diet and high energy low protein diet differs minimally in nutritient levels of MJ/kg (11,03 vs 12,75 MJ/kg), and crude protein (16,2 % vs 13 %) 

What is the difference between ALT and GPT ?

More information about PQQ in nutritiens would be interesting.Comprehensive figure of the potential effects of PQQ would help to understand better its role.

The effect of fructose was not analysed. 

Discussion is too long, and not focusing on PQQ role, decriptive like a biochemical textbook.

Author Response

NAFLD prognosis is depend on NASH development and liver fibrosis progression. I miss histological analysis of the liver to determine inflammatory activity -such as NASH SAF score in humans.

Response: As the reviewer said, NAS and SAF scores are widely used in humans to comprehensively evaluate liver injury and liver fibrosis for prognosis of patients with fatty liver. Based on NAS or SAF scores, NAFLD is recommended to classify into simple fatty liver, early NASH (F0, F1), fibrous NASH (F2, F3), and NASH cirrhosis. In this study, the NAFLD chick model induced by a high-fat-diet for only one month is belonging to simple fatty liver before the early stage of NASH. Therefore, NAS and SAF scores were not used in the present study for NAFLD prognosis.

The diet period was too short. Only one month. Control diet and high energy low protein diet differs minimally in nutrient levels of MJ/kg (11,03 vs 12,75 MJ/kg), and crude protein (16,2 % vs 13 %) 

Response: For laying hens during the laying peak with egg production rate of 93.3%, it is in fact a big challenge to increase ME content to 12.75 MJ/kg and decrease crude protein level to 13.0% in the diet. At the end of 4-week trial, the hens fed the high-energy low-protein diet showed obvious symptoms of fatty liver with significantly higher crude fat and TG content and more fat vacuoles relative to the control group. Therefore, the experimental period and diet formula in the present study are appropriate and successful for constructing a NAFLD chick model to test the effects of PQQ on fatty livers.

What is the difference between ALT and GPT?

Response: ALT is also called GPT, and AST is also called GOT. It’s a mistake in our manuscript; we have corrected GPT into GGT in the manuscript. In addition, we use AST instead of its old name, GOT.

More information about PQQ in nutrition would be interesting. Comprehensive figure of the potential effects of PQQ would help to understand better its role.

Response: In the Introduction part, as the reviewer suggested, we supplemented related information about nutritional roles of PQQ, before to introduce its antioxidant functions. The supplemental content is as follows in Line 63-65 of the revised manuscript.

PQQ has a wide range of nutritional functions, such as promoting microbial and plant growth, and being an essential nutrient for animals and effecting the growth, reproduction, and immune.

The effect of fructose was not analyzed. 

Response: The nutritional role of PQQ has been vastly demonstrated as a cofactor in various enzymes including methanol dehydrogenase, glucose dehydrogenase, alcohol dehydrogenase, aldehyde dehydrogenase. In this study, we mainly focus on the antioxidant function of PQQ, not its well acknowledged nutritional functions. Therefore, the effect of PQQ on fructose metabolism was not analyzed.

Discussion is too long, and not focusing on PQQ role, descriptive like a biochemical textbook.

Response: According to the reviewer’s suggestion, we simplified or deleted the introduction of common sense in the Discussion part, such as the meaning of indexes LDL, HDL, TG, TC, ALT, AST, CS, mtDNA, PGC-1α, NRFs, TFAM, CAT, GSH, ATPase, etc. Made the discussion more focused on the PQQ function.

Reviewer 3 Report

Although of potential interest, the article lacks many procedures although the results have been reported (GSH, CAT,....histology,...). This enable the reviewer to perform a complete evaluation of the article

Author Response

Although of potential interest, the article lacks many procedures although the results have been reported (GSH, CAT,....histology,...). This enable the reviewer to perform a complete evaluation of the article. The method section is too concise and lacks reference for the used methods. A number of procedure are not cited but presented in the results (GSH, CAT,....histology,...). The authors must describe in details the used methods for all the evaluations reported. 

Response: As the reviewer suggested, we improved the materials and methods part.

Firstly, we supplemented the detailed description of procedures for histological analysis in Line 376-382 of the revised manuscript as follows.

4.4. Histological analysis of the liver

Fixed liver tissues were dehydrated and then embedded in paraffin. Carved wax blocks were cut into serial 4-mm-thick sections using a slicer (A550, MEDITE, Burgdorf, Germany). Then, the sections were dewaxed and stained with H&E. Five pathological sections per sample were observed using a light microscope (CK-40, Olympus, Tokyo, Japan) at 40 × magnification. The representative sections with the average level in each treatment were selected for histological comparison.

Secondly, in Line 445-461 of the revised manuscript, we supplemented and improved the methods for detecting the activity of CAT, CS, CCO, and AST, and GSH content in the Chemical Analysis section. Besides, the reference for the used methods was cited. The revised content is as follows.

Liver tissue samples were vacuum freeze-dried and then ground in liquid nitrogen. The crude fat content of livers was determined by ether extraction. For liver homogenates and cell lysates, the contents of TG, TC, HDL-C, and LDL-C were determined by the glycerol phosphate oxidase (GPO-PAP) method [63] using commercial kits. The activity of T-SOD was evaluated by the xanthine oxidase-NBT (nitroblue tetrazolium) method [64] using a commercial kit. The above kits were purchased from Beihua Kangtai Clinical Reagent Co. LTD (Beijing, China). The MDA content was measured by the thiobarbituric acid (TBA) method [65] using a commercial kit. The colorimetric (CI) method was used to determine the activity of ALT, AST, ATPase, CAT, CCO, ChE, CS, and LDH using commercial kits and a multi-detection microplate reader (Synergy 4, BioTek, VT, USA ). Besides, the GSH and ROS contents were also detected by the CI method using commercial kits. The above kits were purchased from Nanjing Jiancheng Bioengineering Institute (Nanjing, China). Protein concentrations of cell lysates were determined using a BCA Protein Assay Kit (Huaxingbio Science, Beijing, China). All the procedures were conducted strictly in accordance to operation instructions of kits.

  1. Bucolo G.; David H. Quantitative determination of serum triglycerides by the use of enzymes. Clin. Chem., 1973, 19, 476–482.
  2. Naoghare, PK.; Kwon, HT.; Song, JM. On-chip assay for determining the inhibitory effects and modes of action of drugs against xanthine oxidase. J. Pharm. Biomed. Anal., 2010, 51, 1-6.
  3. Ulu, H. Evaluation of three 2-thiobarbituric acid methods for the measurement of lipid oxidation in various meats and meat products. Meat Sci. 2004, 67, 683-687.

Round 2

Reviewer 1 Report

The text has been improved but the experiments suggested have not been performed. The literature cited does not compensate for their absence, but rather supports its need to validate the model.

Reviewer 3 Report

The methods reported by the authors for measuring GSH, ROS and other assay are not validated for research purposes